# Modeling of Vessel Traffic Flow for Waterway Design–Port of Świnoujście Case Study

**Agnieszka Nowy [1], Kinga Łazuga [1] , Lucjan Gucma [1], Andrej Androjna [2] , Marko Perkovič [2] and Jure Srše [2,]***

1. Faculty of Navigation, Maritime University of Szczecin, 70-500 Szczecin, Poland; a.nowy@am.szczecin.pl (A.N.); k.lazuga@am.szczecin.pl (K.Ł.); l.gucma@am.szczecin.pl (L.G.)
2. Faculty of Maritime Studies and Transport, University of Ljubljana, 6320 Portorož, Slovenia; andrej.androjna@fpp.uni-lj.si (A.A.); marko.perkovic@fpp.uni-lj.si (M.P.)
* Correspondence: jure.srse@fpp.uni-lj.si

**Abstract:** The paper presents an analysis of ship traffic using the port of Świnoujście and the problems associated with modelling vessel traffic flows. Navigation patterns were studied using the Automatic Identification System (AIS); an analysis of vessel traffic was performed with statistical methods using historical data; and the paper presents probabilistic models of the spatial distribution of vessel traffic and its parameters. The factors that influence the spatial distribution were considered to be the types of vessels, dimensions, and distances to hazards. The results show a correlation between the standard deviation of the traffic flow, the vessel sizes, and the distance to the hazard. These can be used in practice to determine the safety of navigation and the design of non-existing waterways and to create a general model of vessel traffic flow. The creation of the practical applications is intended to improve navigation efficiency, safety, and risk analysis in any particular area.

**Keywords:** vessel traffic flow; the safety of navigation; probabilistic model; Świnoujście approach

## 1. Introduction

In the context of high and increasing interest in maritime space for different purposes, the establishment of new areas of maritime infrastructure or ecologically specific areas will be necessary. Such interventions will also lead to changes in the nature and parameters of shipping routes. A tool is therefore needed to analyse the impact of changes in maritime space. The study of shipping traffic flows is crucial to the safety and efficiency of navigation. Analysis of vessel traffic helps to understand ships' navigation patterns and provides the necessary input for maritime risk assessment tools. Specific models need to be created for selected areas. A general model of vessel traffic flows for the Baltic Sea is the goal of the studies.

The studies on risk analysis have enhanced the work on traffic flows. The first studies were based on a random spatial distribution of ships with no distinction between ship types [1,2]. A normal and uniform distribution over coastal areas, vessels, and navigational infrastructure characteristics was used as the theoretical distribution [3]. Vessel distribution adjustment studies were conducted for coastal areas, necessitated by limited access to data [4]. The authors proposed to calculate the probability of collision or grounding by defining the lateral limits of the navigable area in a Weibull-, Rayleigh-, or Gaussian-type distribution.

Expert studies on a group of navigators, based on radar data analysis, were used to conduct offshore collision risk studies. It has been shown that the distribution of ship distance from the average route depends mainly on the length of the route [5]. A linear model of ship positions on track was constructed.

Consideration of more independent data for the model was used by Pedersen [6]. Vessel traffic was classified according to vessel type, displacement, length, whether it was loaded or ballasted, ship speed, draught, and ice-class. The author created a model

to calculate the collision risk in busy shipping lanes and studied the distribution of the different traffic classes.

Currently, AIS data are used in studies of actual ship behaviour. In recent years, numerous studies have been conducted to assess collision risk and traffic. A model presented by Goerlandt and Kujala [7] is based on an extensive dynamic microsimulation of vessel traffic using the Monte Carlo simulation. Inputs to this model include routes, number of vessels on each route, departure times, ship dimensions, and speed. Detailed studies of vessel traffic statistics were conducted.

Since the traffic looks different in different maritime areas, the studies are conducted to determine the parameters of traffic flows for specific waters. Studies on shipping traffic flow have been conducted in the Baltic Sea [8,9], in the Japan Strait [10], in the Adriatic Sea [11], in the port of Lisbon [12], off the coast of India [13], in the Singapore Strait [14]. and in China: the Dagusha Channel [15] and the Ningbo-Zhoushan Port [16]. Two comprehensive studies on traffic analysis in the Istanbul Strait have been published [17,18]. The statistical analysis of vessel traffic and the collision risk model off the coast of Portugal have been developed [19].

A classical traffic flow theory was used in an initially developed mathematical model [20]. A back propagation (BP) neural network was used to predict vessel traffic flow [21]. Forecasting models of vessel traffic flow using a radial basis function (RBF) neural network, grey forecasting, auto-regression, and a combined model with support vector machines (SVM) were presented by Wang [22]. A novel method for modelling maritime traffic based on the concept of potential field was created by [23], where the fluctuations of traffic relative to time are studied. Automatic detection of traffic flow based on kernel density estimation is proposed by [24].

The paper presents studies of traffic flow in the approach to the port Świnoujście as a part of research into a general mathematical model of vessel traffic streams. AIS data are used.

## 2. Methodology

### 2.1. Methods Used

The methods presented here are directed toward finding the relation between the standard deviation of ship position with waterway length and width. The variability in the route planning by navigators results mainly from their individual preferences, mainly concerning distances to the dangers. With the assumption that the route consists of the straight leg of length $L_{tr}$, its variability is the result of the start and end waypoint selection. The navigator's choice of given waypoints can be treated as random, so it can be described utilizing one- or two-dimensional random variables, as shown in Figure 1. A variable describing the starting and endpoint of the route can be determined through the expert opinion carried out by a group of navigators or by radar or AIS observations. Parameters of distributions depend especially on water area type, availability of position fixing systems, size, type of ship, and length of route. Studies of the available literature were carried out by Haugen [5], based on expert opinions, and his observations and models showed that the distribution of the distance of ships from the average route depends especially on the route's length. This led to the linear model of standard deviation dependence from the route length for routes up to 600 NMs of length in the following form:

$$\sigma_{tr} = -0.027 + 0.0073 L_{tr} \tag{1}$$

where:

- $\sigma_{tr}$ is the total standard deviation from the mean route due to the route planning, means of ship control, and external factors;
- $L_{tr}$ is the length of the route (NM).

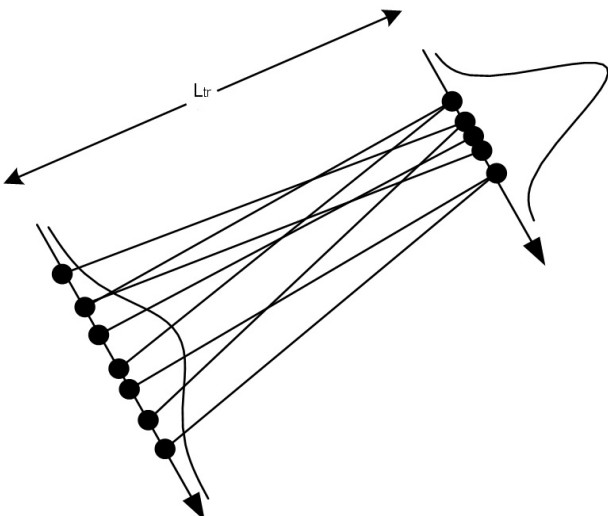

**Figure 1.** Influence of the distribution of initial and final points of the ship's route.

The standard deviation obtained in such a way enables us to estimate the geometrical probability by a normal distribution.

Fuji [25] introduced dependence of the geometrical probability estimation in the following form:

$$P_G = \frac{D_p + B}{D} \tag{2}$$

where:

- $D_p$ is the available width between the bridge piers'
- $B$ is the the breadth of the ship;
- $D$ is the width of the waterway.

The above formula is very simplified and assumes the uniform distribution of ship positions in the waterway. McDuff's [2] studies enabled evaluation of simplified relations for the determination of geometrical probability in the following form:

$$P_G = \frac{4l_s}{\pi D} \tag{3}$$

where:

- ○ $l_s$ is the distance to stop a ship;
- ○ $D$ is the available waterway width.

Kuroda's [26] research enabled determination of the relations between the parameters of normal distribution and the position of a ship on the waterway, as follows:

$$m = aD; \quad \sigma = 0.105D \tag{4}$$

where:

- $m$, $\sigma$ are the normal distribution parameters;
- $D$ is the available width of the waterway;
- $a$ is the parameter assumed as 0.2 for canals with central marking and 0.1 without such marking.

AASHTO [27] is the recommended application for the normal distribution of parameters:

$$m = 0; \quad \sigma = L \tag{5}$$

where:

- $m$, $\sigma$ are the normal distribution parameters;

- *L* is the length of the ship.

The calculations are partially performed using the mathematical software tool IWRAP MK2 [28], recommended by the International Association of Marine Aids to Navigation and Lighthouse Authorities (IALA). The main objective of the research was to search for the dependencies between the main parameters of the ship traffic stream in the waterway system. Such dependencies are necessary to build statistical models aiming at a generalization of ship traffic and presenting it in the form of useful models. Therefore, the article has a strong utilitarian character. The authors use such models in later analyses when they need to design anticipated waterway features. In the paper, apart from the typical study of statistical distributions, which can be used to model the ship's position in relation to the track axis, a number of other dependencies were sought, such as the dependence of the standard deviation of the ship's position in relation to the length of the section, the dependence of the ship's size, including draught and width, on the position on the fairway, or such important dependencies of the mean and standard deviation of the ship's position in relation to the distance to safe isobaths. Such models are an invaluable tool in waterway design because they use the standard deviation of ship positions, and therefore, it is possible to determine the probability of accidents in non-yet-existent marine traffic engineering systems.

### 2.2. Analyzed Area

Located in north-western Poland at the mouth of the Oder River, the universal ports of Szczecin and Świnoujście are among the most extensive port facilities in the Baltic Sea region. Świnoujście offers dense direct traffic to/from Sweden, thanks to the largest ferry terminal in Poland; the port also specializes in handling various bulk cargoes. The port areas are overseen by vessel traffic service systems, defined by lines marking the reporting obligation via VHF radio.

Marine traffic in the approach to Świnoujście consists of two main types: transit and cross-strait traffic. The first is the traffic of cargo and passengers ships transiting to the ports Świnoujście and Szczecin. The second is the traffic of small passenger vessels and pleasure boats operating on the routes between the small ports. This analysis was based only on the latter type.

The authors analysed the movement of ships in the main flows from Arkona, as shown in Figure 2a,b. The daily average of ship passages using these lanes (towards Świnoujście) is about 25, including ferries, passenger ships, and fishing boats.

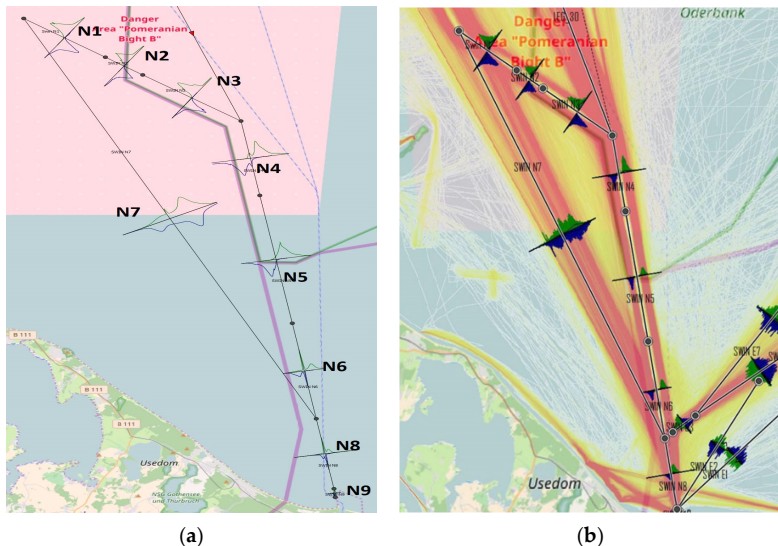

(**a**)   (**b**)

**Figure 2.** Pomeranian Bayresearch area: (**a**) Established sections; (**b**) density plot.

The traffic streams in the Pomeranian Bay were divided into 9 sections, called legs, as shown in Figure 2a. Section N1 is located on the north side of the northern approach fairway near buoy N1, while N9 is the section near the head of the Świnoujście breakwater (buoy A). Section N7 represents the western approach to the port. The sections have been defined so that the internal navigation conditions within each section are constant (e.g., distance to isobaths, aids to navigation).

### 2.3. Data

The studies were conducted based on raw data (National Marine Electronics Association NMEA records) from AIS obtained from the Polish Maritime Administration. Vessel traffic was analysed using data from March to May 2019. The three most common vessel groups, general cargo (GC), oil product tankers (OPT), and passenger vessels (PAS), were considered the most frequent in a given area. Ships longer than 50 m were included in the study. AIS raw data were processed using the IWRAP MK2 application [25], a modelling tool recommended by IALA for maritime risk assessment. The traffic patterns in Figure 2b are shown in a density plot, which helps to identify the location of shipping routes. Statistical analysis was performed using Statistica 10.0.

A cross-section of the leg was made, and a histogram was prepared for each direction; the mathematical representation was prepared with various probability functions. The data were divided into classes of vessels, and each crossing line was examined. The study mainly consisted of matching the distribution of traffic with respect to the axis and obtaining the mean and standard deviation of the lane for three groups of vessels. Traffic characteristics, such as speed and course distribution, ship types, and dimensions, were determined. The main parameters of the traffic stream and their changes through the subsequent sections/legs were analysed.

The ship's crossings show the actual behaviour of navigators and determine the influencing parameters on vessel routes. The correlation between the mean, standard deviation of spatial distribution, and distance to safe isobaths, leg length, width [*B*], and ships' draught [*T*] were verified.

### 3. Results

The preliminary analysis of vessel traffic was based on a traffic density map. After the analysis, it was found that, in some cases, the main traffic flows were composed of two characteristic vessel traffic flows that could be selected and analysed in detail, as shown in Figure 3a. One flow consisted of vessels entering the port using the fairway marked by navigational aids; the second contained vessels passing outside the defined fairway. All regression models presented in this paper refer to the second stream, for inbound vessels only.

No factor influencing the navigator's route selection was identified in studies. It can be inferred that it is random or due to weather conditions, which was not considered in the study. It can only be concluded that vessels with a draught greater than 10 m follow the main fairway, as shown in Figure 3b. It can be seen that vessels enter and leave virtually the entire length of the fairway. For this reason, the number of vessels at successive gates on the same lane varies.

### 3.1. Ship Spatial Distribution

Figure 4 shows the fitted functions of the spatial distribution of ship positions for three groups of ships. The graphs show that most ships navigated on the starboard side of the fairway, which is the standard behaviour for ships.

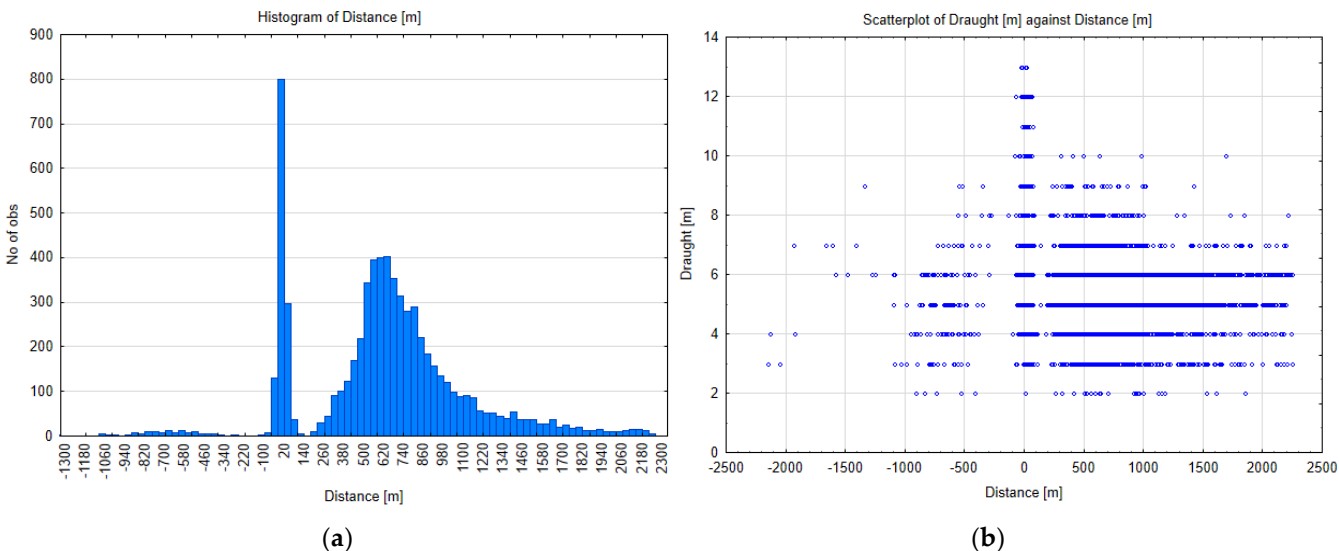

**Figure 3.** (**a**) Spatial distribution of ship's position relative to track axis at leg N4, incoming traffic; (**b**) scatterplot of ships draught versus distance from centre.

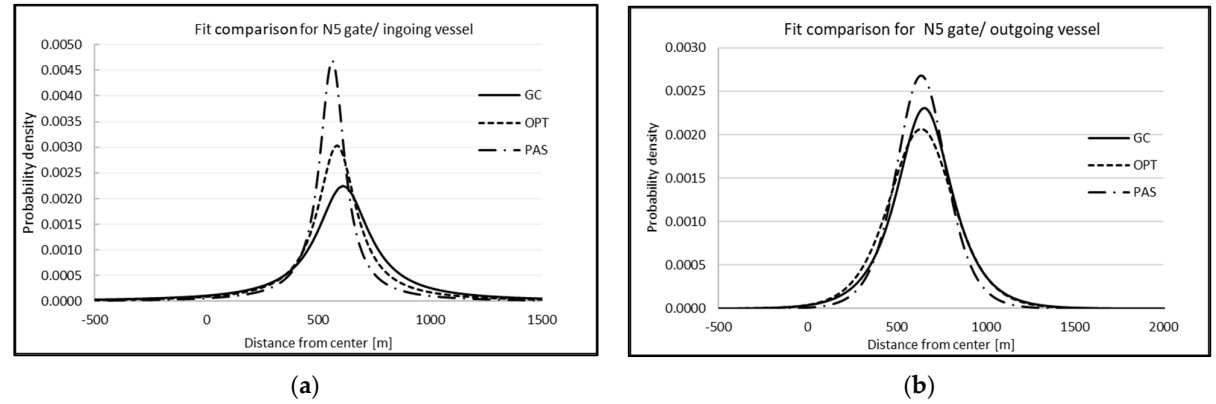

**Figure 4.** (**a**) Spatial distribution of ship hull position relative to track axis at cross-section N5 for three ship types, outside the marked fairway for inbound and (**b**) outbound ships.

For the subsequent section of the route, the spatial distribution of ship position relative to the centreline of the traffic route was determined. Figure 5a shows the mean and Figure 5b the standard deviation of vessel position from cross-section N1 to N9. For the gates numbered N4 to N6, the traffic flow outside the marked and dredged channel was considered. The graphs show that the oil product tankers navigated closer to the centreline of the fairway than general cargo vessels. The mean and standard deviation had lower values. Like the vessels on the regular line (which allowed for following the same defined route), the passenger vessels travelled the closest to the centreline of the defined route. The traffic flow parameters for all groups of vessels decreased relatively by the same value when approaching the port entrance (leg N9). The difference in the traffic flow parameters of passenger vessels resulted from the character of the traffic. The vessels mostly joined the traffic at gates N3 and N4 and steer as close as possible to the marked channel.

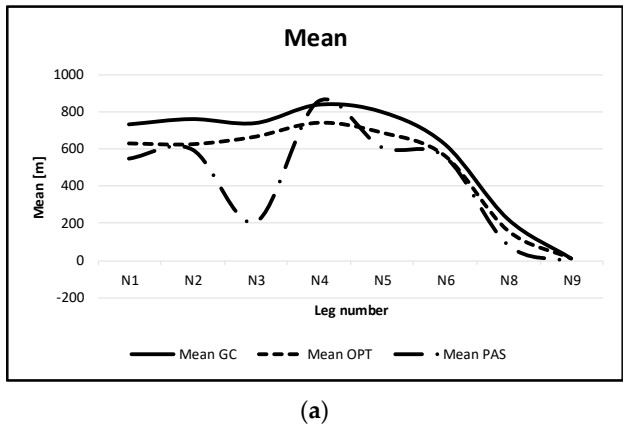 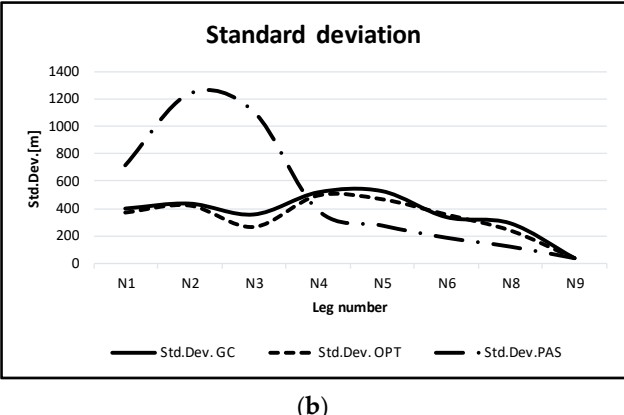

(**a**)  (**b**)

**Figure 5.** (**a**) The mean and (**b**) standard deviations of ship position for consecutive legs N1 to N9.

### 3.2. Average Speed in the Fairway

Vessel speeds are constantly changing in the approaching fairway. The average speed for each type of vessel at each crossing gate was calculated. As can be seen in Figure 6a, the average speed varied between 8 kn and 14 kn. The speeds of OPT and GC vessels are comparable, but significantly higher speed values can be seen for the PAS vessel. The different speed selection for different vessel classes resulted from the manoeuvring characteristics of the vessels. Passenger ships (ferries) had excellent manoeuvrability. Compared to GC or OPT ships, it took less time to reduce speed. The second factor is that the navigators on PAS ships were familiar with local waters and special conditions and could guide the ships through busy areas. The masters were Pilotage Exemption Certificate owners, so there was no reason to reduce the speed for pilot embarkation.

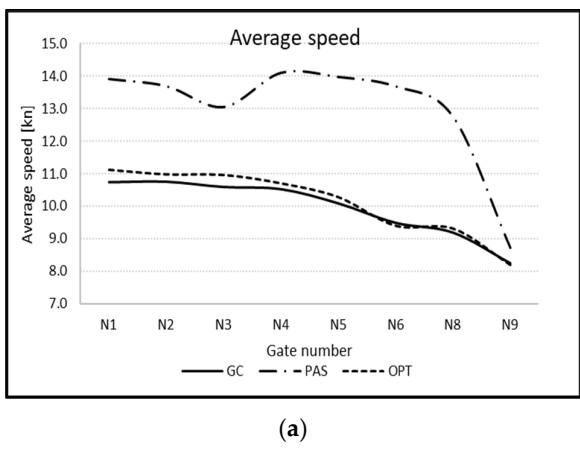 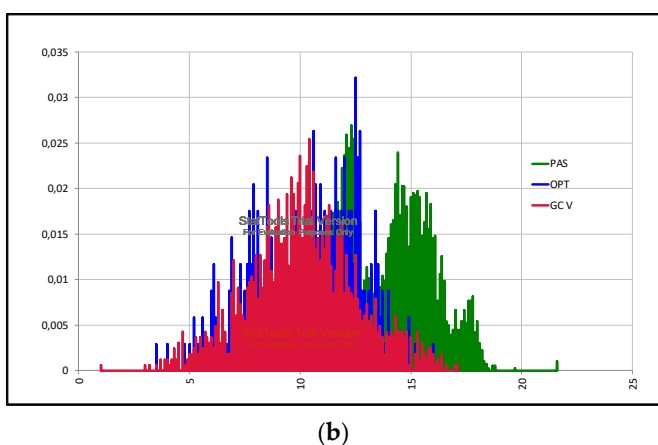

(**a**)  (**b**)

**Figure 6.** (**a**) Average speed of an incoming vessel; (**b**) Histogram of vessel speeds on N5 for three groups of ships.

Figure 6b shows the histogram of ship speed for three groups of ships on the N5 leg. The speed varied from 5 kn to 18 kn. Only a few ships navigated at a speed of less than 5 kn or more than 18 kn. Most vessels do not navigate at full speed, and vessels can be expected to change speed during the passage depending on the situation. Vessels with draughts $T$ greater than 10 m navigate at a reduced speed in the range of 4 kn–8 kn, as shown in Figure 7.

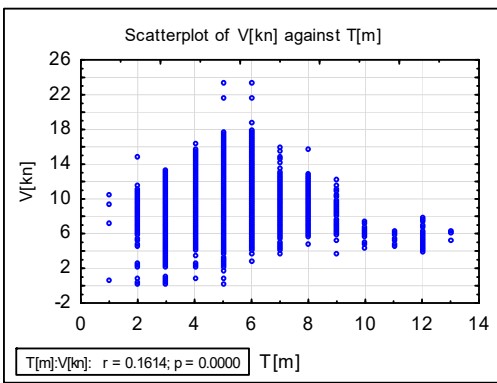

**Figure 7.** The scatterplot of vessel speed against draught.

*3.3. Regression Analysis of Traffic Flow Parameters*

3.3.1. The Model Including Draught T

To construct regression models of marine traffic flow, all passages for consecutive gates were considered. It can be observed that there was a difference between the mean and standard deviation of vessel position from the centre of the route for selected groups of vessels at different gates. It was decided to make these groups permeable and see if there would be any statistically significant difference in the results. To compare three independent groups with respect to the quotient variables, a nonparametric Kruskal-Wallis test was applied. The Kruskal-Wallis test could cover the null hypothesis for three or more samples.

The null hypothesis states that all samples come from the same distribution. In this case, continuous distributions were assumed. On the other hand, if it can be assumed that all population distributions have the same shape (normal or not), the hypothesis states that the medians of the populations are identical [29].

The H-statistic for the Kruskal- Wallis Test is calculated using the defined formula:

$$H = \frac{1}{C}\left(\frac{12}{N(N+1)}\sum_{j=1}^{k}\left(\frac{\left(\sum_{i=1}^{n_j}R_{ij}\right)^2}{n_j}\right) - 3(N+1)\right) \tag{6}$$

where:

- $H$ is the the value of Kurskal–Wallis test;
- $N = \sum_{j=1}^{k}n_j$;
- $N$ is the number of all observations;
- $k$ is the number of compared groups;
- $n_j$ is the sample size for $(j = 1, 2, \ldots, k)$;
- $R_{ij}$ is the the rank assigned to the value of the variable, for $(i = 1,2, \ldots nj),(j = 1,2, \ldots, k)$;
- $C = 1 - \frac{\sum(t^3 - t)}{N^3 - N}$ is the adjustment factor for ties, t-number of cases included in tied rank.

The H statistic has an asymptotic (for a large sample size) distribution of the chi-square with the number of degrees of freedom determined by the formula: *df = (k − 1)*. The next step is to compare the H-value with the critical chi-square value $\chi^2_{0.05}$ for *df*. If the critical chi-squared value is less than the H statistic, we can reject the null hypothesis that the medians are equal. If the chi-squared value is not less than the H-statistic, there is insufficient evidence that the medians are unequal.

The Kruskal-Wallis test indicates significant differences between categories but does not specifically say which categories differ, so a post-hoc test was performed with the Bonferroni correction. The Bonferroni correction is an adjustment to *p*-values made when multiple statistical tests are run simultaneously on the same data.

The test showed that the samples for legs N1 to N6 were from the same distribution, so there was no significant difference between the samples. For samples N7, N8, and N9, the null hypothesis was rejected. The results presented in the remainder of this paper are for gates N1–N6.

Figure 8a shows the relationship between the mean distance of ships from the centre and the draught *T* of the ship. Simple linear regression models were constructed for three groups of ships. It can be seen that there was a linear relationship between these parameters, with a coefficient of determination of 0.8 for GC, 0.7 for OPT, and 0.27 for PAS. For the GC and OPT vessels, the regression coefficient was satisfactory according to the accepted interpretation. It can be concluded that the higher the value of the draught, the closer to the middle of the track the navigator is sailing. The correlation between standard deviation and draught of the vessel was lower than for the mean value. The best coefficient of determination was for PAS ($R^2 = 0.60$), as shown in Table 1. Although the coefficient of determination for the GC group was low, the independent parameter *T* was statistically significant for the model ($p = 0.004$). For OPT vessels, the parameter *T* was not statistically significant ($p > 0.05$). An interesting result is a correlation between standard deviation and draught for passenger vessels. As the draft increased, the standard deviation increased accordingly. These results confirmed the specificity of passenger ferry traffic in the selected area, and further analysis is required. The model is valid for PAS ships with $T \geq 4$ m.

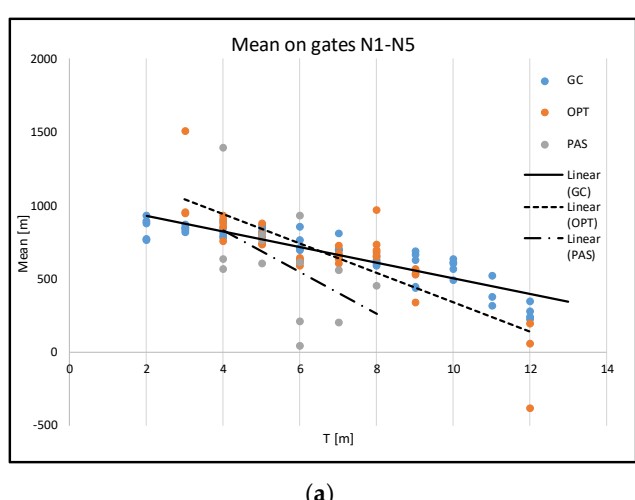

(**a**)

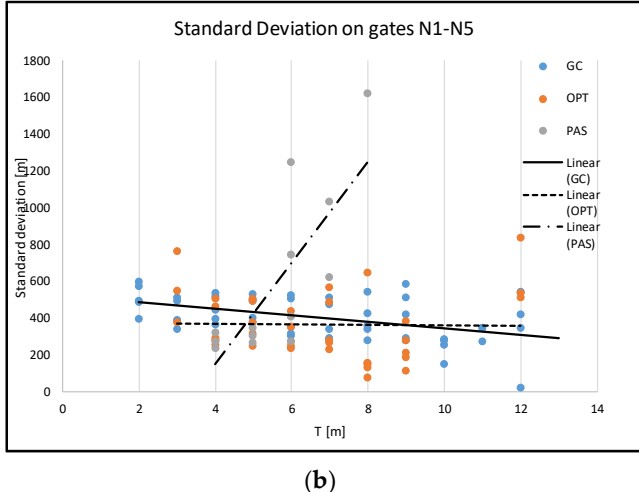

(**b**)

**Figure 8.** (**a**) The mean of the ship's distance from the centre; (**b**) standard deviation of ship distance from the centre to the ship's draught.

**Table 1.** The parameter of the simple regression T-model.

| Dependent Variable | Ships Type | Coefficient Intercept | Coefficient T[m] | $R^2$ | $F$ | $p$ |
|---|---|---|---|---|---|---|
| Mean | GC | 1037.7 * | −53.8 * | 0.7993 * | 208.7330 * | 0.0000 * |
| Mean | OPT | 1340.6 * | −100.1 * | 0.6801 * | 72.2755 * | 0.0000 * |
| Mean | PAS | 1367.1 * | −136.8 | 0.2667 | 3.5840 | 0.0849 |
| Std.Dev. | GC | 523.9 * | −18.1 * | 0.2265 * | 14.3461 * | 0.0004 * |
| Std.Dev. | OPT | 372.4 * | −1.2 | 0.0003 | 0.0093 | 0.9238 |
| Std.Dev. | PAS | −663.1 | 224.1 * | 0.6101 * | 17.2121 * | 0.0016 * |

Legend: * statistical significance to regression parameter.

### 3.3.2. Model Including Width B

Testing the correlation between mean, standard deviation, and width of the ships showed that a regression model could only be constructed for the mean width relation

for GC and OPT ships (for gates N1N6). The wider the ship, the closer to the middle of the track the navigator sails, as shown in Figure 9. $R^2$ for the GC was 0.37, which means that the model explained 37% of the variability of the response data around its mean, as shown in Table 2. For OPT vessels, the model explained 45%. Although the coefficient of determination was not high, the statistic $p$ was equal to 0.00. For both models, the significant $p$-value indicates that we can reject the null hypothesis that the coefficient is equal to zero (which still indicates a fundamental relationship between the significant predictors and the response variable). The coefficient of determination may be low because the model predicts the behaviour of the navigator. People are more complex, thus less predictable than, for example, physical processes.

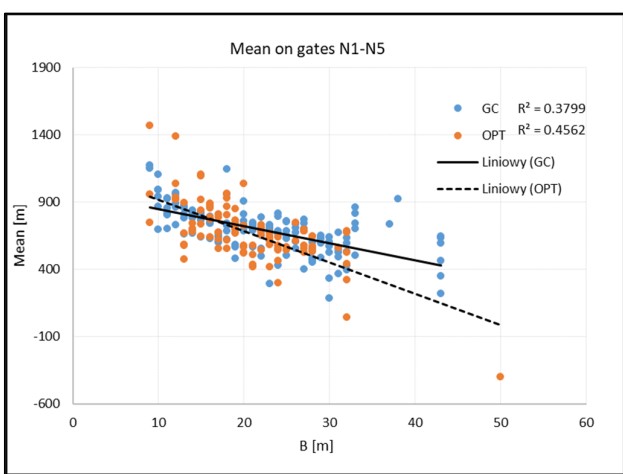

**Figure 9.** Mean of ships distance from the centre to the width of the ship.

**Table 2.** The parameter of simple regression B-model.

| Dependent Variable | Ships Type | Coefficient Intercept | Coefficient B[m] | $R^2$ | F | p |
|---|---|---|---|---|---|---|
| Mean | GC | 973.2 * | −12.7 * | 0.3799 * | 90.6729 * | 0.0000 * |
| Mean | OPT | 1150.6 * | −23.3 * | 0.4562 * | 83.8872 * | 0.0000 * |
| Mean | PAS | −6.2 | 23.9 | 0.0245 | 0.5532 | 0.4649 |
| Std.Dev. | GC | 398.7 * | −0.7 | 0.0015 | 0.2211 | 0.6389 |
| Std.Dev. | OPT | 304.2 * | 0.1 | 0.0000 | 0.0013 | 0.9715 |
| Std.Dev. | PAS | 0.8 | 42.3 | 0.0000 | 0.0003 | 0.9857 |

Legend: * statistical significance to regression parameter.

### 3.3.3. Model Including Length of the Leg $L_{tr}$

The authors analysed the correlation between the standard deviation of the ship's position from the centre of the track and the length of the leg. Predictor $L_{tr}$ had a $p$-value of 0.0065, which shows the statistical significance of the achieved results. The coefficient of determination $R^2 = 0.42$ indicates that the model explained 42% of the variability in the response data around its mean. This led to the interesting conclusion that the standard deviation of ship position in traffic flow is higher when the length of the route increases, as shown in Figure 10. The research follows Haugen's research, indicating that the distribution of the distance of vessel from the average route depends mainly on the length of the route [5]. The Haugen model is applicable for routes up to 600 Nm in length.

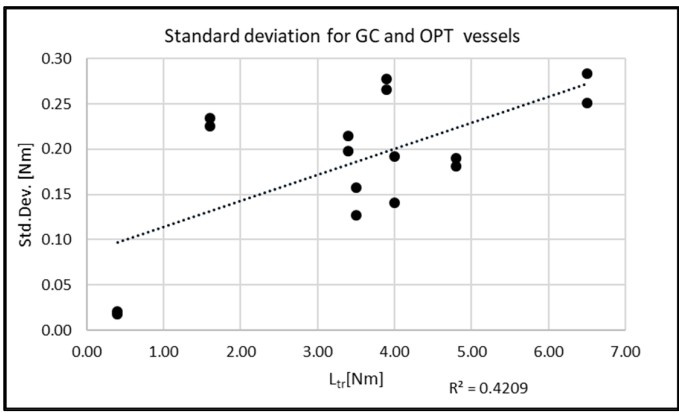

$$SD = 0.0852 + 0.0287*L_{tr}$$

**Figure 10.** The standard deviation of the ship's distance from the centre to the length of the leg.

### 3.3.4. Model Including Distance to Isobaths $D_{10m}$

The parameters of the traffic flow changed with the distance from the hazard. In the studied area, the correlation was defined for the distance to 10 m isobaths on the starboard side $D_{10m}$. As the distance to the defined isobaths increased, the spatial parameters of the traffic flow increased, as shown in Figure 11a,b. This behaviour is understandable. The more complex (the narrower) the area is for navigation, the more accurate the steering of the vessel.

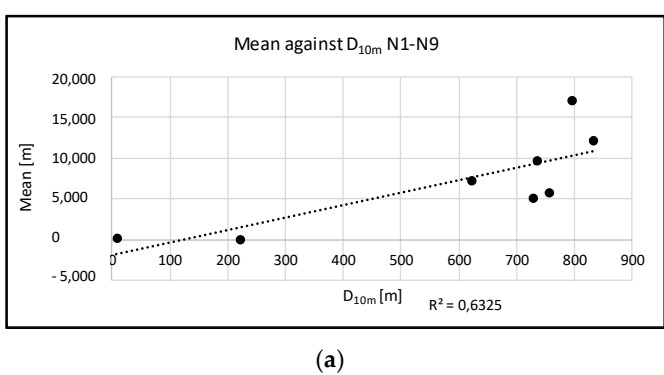

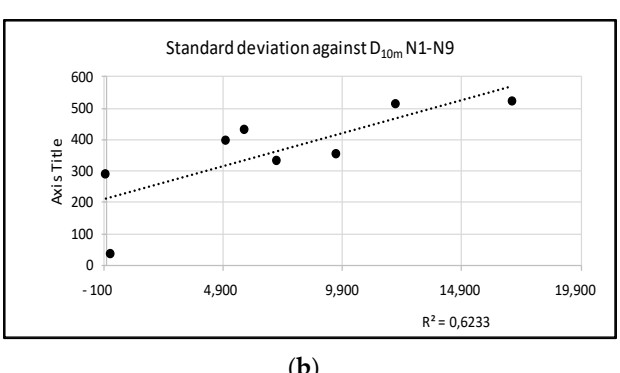

(**a**)                                                 (**b**)

**Figure 11.** (**a**) Mean versus distance to 10 m isobaths $D_{10m}$, $R^2 = 0.63$, $p = 0.0183$, $F = 10.3245$; Standard Deviation = 292.23 + 0.0416 * distance to isobaths. (**b**) Standard deviation to distance, $R^2 = 0.62$, $p = 0.0197$, $F = 9.927$; Mean = 212.3691 + 0.0210 * distance to isobaths.

### 3.4. Influence of the Ship's Draught and Width

It is a natural behaviour of mariners to navigate more cautiously on ships with less manoeuvrability. On the scatterplots in Figure 12a,b, it can be seen that, as *T* increased, the distance of ships from the centre of the track decreased (for *T* > 7 m rapidly). For vessels with more than 8 m draught, the steering is more precise as the vessel follows the given track from the beginning of the fairway, as shown in Figure 12b. Ships with a shallower draught can join the main stream in this section (N8). The case studied location was a straight fairway, where vessels do not normally need to alter their course during passage. However, it should be noted that the approach channel was relatively narrow, so there was less opportunity to change course, and doing so was likely only during a perceived emergency.

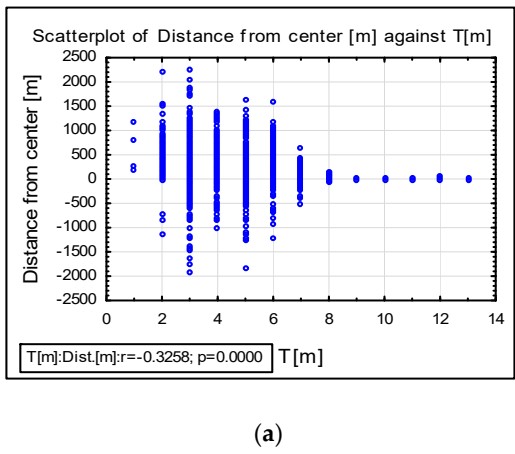

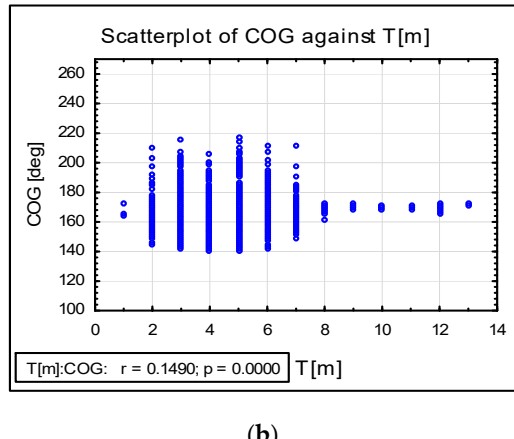

(**a**) (**b**)

**Figure 12.** (**a**) Scatterplot of distance from centre versus draught *T*. (**b**) Scatterplot of COG versus draught *T*.

The wider the ship, the closer to the centre of the track she navigates, as shown in Figure 13a. The parameter *B* of the ships particulars had a similar effect to *T*. The wider the ship, the more accurately she sails. The deviation from COG was much less, as shown in Figure 13b. The coefficient of determination was low, but the *p*-value was 0.000, indicating that parameter *B* was statistically significant.

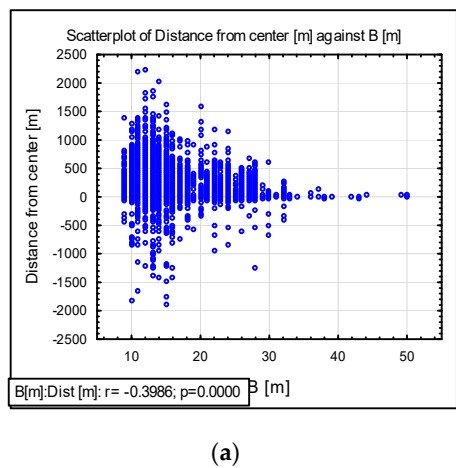

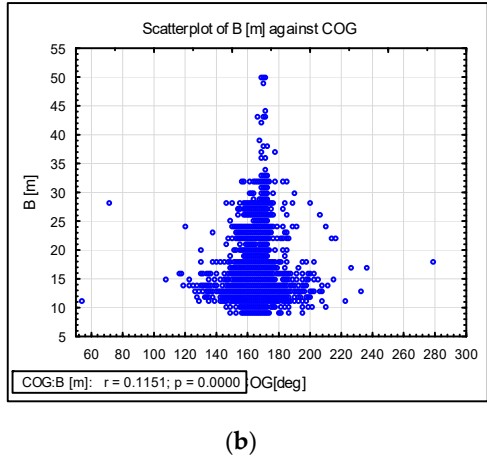

(**a**) (**b**)

**Figure 13.** (**a**) Scatterplot of distance from centre versus draught *T*. (**b**) Scatterplot of a vessel's width versus COG.

## 4. Conclusions and Discussion

The researchers present AIS data analysis for the approach to Świnoujście to gain better insights into the behaviour and manoeuvring of ships. The defined distribution of the ship's position, speed, courses, dimensions of vessels, and the relationship between these factors form the basis for the mathematical model of traffic flow. Some simple regression models describing the behaviour of the vessels have been presented.

The significant predictor of vessel traffic flow parameters in the model was vessel draught *T*. For GC vessels, 80% of the variation in the variable "mean" distance to the centre of the track was explained by the model: 68% for OPT; 27% for PAS ships. However, in this case, the predictor was not significant. For the dependent variable "standard deviation", the model explained 23% of the variation for GC ships, 61% for PAS ships, and 0% for OPT ships. The width of ship *B* was significant only in regard to the parameter GC and OPT ships, where 40% and 45% of the variation using this variable was explained. For the standard deviation, it was not significant. These results suggest that further investigation is necessary in even more restricted (narrower) areas, where the vessel width should have a significant impact.

The authors created two important models for mean and standard deviation of different classes of ship position variability of waterway leg length. This was inspired by Haugen [5] and Japanese studies [1,3,10,26,27]. The validated regression model in the form of $SD = 0.0852 + 0.0287 \times Ltr$ shows a difference from these studies, but the analysed area also differs. Moreover, the authors found that the mean position of ships on the waterway also changed on long waterway legs, which was never discussed before.

Regression was used to compare the standard deviation of vessel position with the length of the track (section). The model explains 42% of the variation for GC and OPT vessels. It is suggested that further studies consider Haugen's studies [5] and the results obtained.

Since the distance to the hazard is a significant factor for the safety of a vessel in traffic, the distance to safe isobaths $D_{10m}$ was analysed using the parameters of vessel traffic flow. The model created explained 63% of the variance for the mean and 63% of the standard deviation. The predictor was significant, with $p < 0.02$.

The authors also studied the influence of ship width and generally ship size on their positions on the waterway. The results turned out to be consistent with previous hypotheses, i.e., larger ships keep closer to the centre of the track but the regression models allowed this to be quantified, which is very important for building general navigation safety models. Similar results were obtained by examining the draught of ships in relation to the safe isobaths.

There are additional factors that contribute to vessel traffic flow that were not considered. These factors include wind, current, visibility, the human factor, and ship interaction. Small vessels of L < 50 m were not considered. In addition, the accuracy of data from AIS should be investigated, and the impact on the model considered.

All waterborne activities lead to changes in the characteristics of vessel traffic flows at sea. In order to calculate the impact on traffic flows, a model of marine traffic flow is required. The developed model enables the design of maritime routes in the development of spatial planning on the seas.

**Author Contributions:** Conceptualization, A.N. and L.G.; methodology, A.N., L.G., K.Ł., A.A., M.P. and J.S.; data collection, A.N. and L.G.; formal analysis, A.N., L.G. and K.Ł.; investigation, A.N., L.G. and K.Ł.; data curation, A.N. and L.G.; visualization, A.N. and L.G.; supervision, L.G., A.A. and M.P; writing—original draft preparation, A.N., L.G., K.Ł., A.A., M.P. and J.S.; internal review, A.A., M.P. and J.S. All authors have read and agreed to the published version of the manuscript.

**Funding:** This research outcome has been achieved under the grant No. 11/MN/IIRM/17 financed from a subsidy of the Polish Ministry of Science and Higher Education for statutory activities. The publication of the paper is also partially financed by the research project (L7-1847; Developing a sustainable model for the growth of the "green port") and the research group (P2-0394; Modelling and simulations in traffic and maritime engineering) at the Faculty of Maritime Studies and Transport, financed by the Slovenian National Research Agency.

**Institutional Review Board Statement:** Not applicable.

**Informed Consent Statement:** Not applicable.

**Data Availability Statement:** The data analysed in this study was a reanalysis of existing data from the Polish AIS data network.

**Acknowledgments:** We thank the Maritime Office in Gdynia for providing the AIS data.

**Conflicts of Interest:** The authors declare no conflict of interest.

## Abbreviations

| | |
|---|---|
| AIS | Automatic Identification System |
| BP | Backpropagation |
| GC | General Cargo |

| IALA | International Association of Marine Aids to Navigation and Lighthouse Authorities |
| IWRAP | IALA Waterway Risk Assessment Program |
| OPT | Oil Product Tankers |
| PAS | Passengers vessel |
| RBF | Radial Basis Function |
| SVM | Support Vector Machines |

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
