# Peer review of "Modeling of Vessel Traffic Flow for Waterway Design–Port of Świnoujście Case Study"

_applsci, doi:10.3390/app11178126_

Round 1

Reviewer 1 Report

It is well written work. Research work were designed property. Results is sound.

The only problem is that writing format and style. It is a little bit redundant. Could you make the writing more concise and highlight the theme and main point of view?

Author Response

Dear Reviewer,

Thank you for your valuable and encouraging comments. We have done our best to consider your suggestion and improve the article. We sincerely hope that we have succeeded in improving our work according to your expectations.

Yours Sincerely,

dr. Andrej Androjna

Reviewer 2 Report

An interesting scientific article deals with the study of ship traffic flow in Pomeranian bay. Using AIS data from March to May 2019, they analyzed the mean and standard deviation of the ship’s position from the center of the analysis area. Based on this, they established different traffic flow models based on simple linear regression depending on the depth of the draft, the width of the ship, the length of the positions, and the distance to the isobath. For the average reader, the article is extremely difficult to read.

Before publishing, I suggest certain minor corrections that would increase the transparency and readability of the article:

- line 200 -: The theory in connection with the Kruskal - Wallis test is presented. It is not clear from the text why and how the test was used. What were the intentions of the authors regarding the use of this? What are the results? How are the results of this test related to the models presented below? Which hypotheses were tested? What is the null hypothesis? The text in this part should be amended, supplemented.

- In Table 1 and Table 2 and somewhere between the text (eg line 286) the symbol "*" is used. What is the purpose of this designation? What does it mean? Needs to be supplemented.

Author Response

Dear Reviewer,

Thank you very much, indeed, for your valuable comments. We have tried to take your suggestion into account in the best possible way to add more value to the article. We sincerely hope that we managed to answer all your respective questions (new article attached). 

Changes/modifications are as follow:

  1. Line 200 comment: »The theory in connection with the Kruskal - Wallis test is presented. It is not clear from the text why and how the test was used. What were the intentions of the authors regarding the use of this? What are the results? How are the results of this test related to the models presented below? Which hypotheses were tested? What is the null hypothesis? The text in this part should be amended, supplemented.

Answer:  The reason of using K-W test is that the test is nonparametric and does not require the normality of distributions of ships positions on the waterway. In this case and also in our other studies most of the distributions are pass statistical tests for normality (for example Shapiro-Wilk test) but not all of them (about 75% is normally distributed in given sections of the waterway). We tried to prove by this test if the samples come from the same distributions. The test shows that the samples for legs N1 to N6 are from the same distribution, so there is no significant difference between the samples. For samples N7, N8 and N9, the null hypothesis was rejected. The results presented in the remainder of this paper are for gates N1-N6. Therefore based on the test result in the later analysis, we joined the samples into one.

  1. In Tables 1 and 2 the symbol "*" is used. What is the purpose of this designation? What does it mean? Needs to be supplemented.

Answer to Rev 2: It shows the statistical significance of a given parameter. We added the description.

Yours Sincerely,

dr. Andrej Androjna

Reviewer 3 Report

Traffic Modelling has drawn much interests by researchers. The reseach topic is interesting in that the traffic flows have statistically anlyzed with the observed AIS  data, and then vessel/trajectory parameters influencing on traffic patterns. 

Regardless of this, the reviewer did not find out any other originality for modelling traffic, except that the authors introducted H-Statistics for a Kruskal-Wallis Test and made regression w.r.t several traffic parameters.   Conclusions should be clarified and described concisely, moving some experimental results to the Section 3 Results.   

This review recommends:

to clarify the purpose of this research and to present the newly proposed one in 1. Introduction. 

to review the existing methods in the Section 2.

to represent the analytic methods and theorectical background on the traffic models in the Section 3 prior to "3. Results".

Author Response

Dear Reviewer,

Thank you very much, indeed, for your encouraging comments. We have tried to take your suggestions into account in the best possible way to add more value to the article. We sincerely hope that we managed to answer all your respective questions (new article attached). 

Changes/modifications are as follow:

  1. »To clarify the purpose of this research and to present the newly proposed one in 1. Introduction.«

Answer: Amended as required at the end of Chapter 1

  1. »To review the existing methods in the Section 2.«

Answer: Some of the methods used we have already described in the previous chapter.

  1. »To represent the analytic methods and theorectical background on the traffic models in the Section 3 prior to "3. Results".«

Answer: Some of the methods used we have already described in Chapter 1.

  1. »the reviewer did not find out any other originality for modelling traffic, except that the authors introduced H-Statistics for a Kruskal-Wallis Test and made regression w.r.t several traffic parameters.

Answer: The novelty according to the authors' opinion is searching for the dependencies between the main parameters of the ship traffic stream in the waterway system. Such dependencies are necessary to build statistical models aiming at the generalization of ship traffic and presenting it in the form of useful models. We agree that the paper has an engineering character but it's positive due to also utilitarian aspect. The authors use such models in later analyses when they need to design waterway features that do not yet exist. They can be useful also to other researchers. In the paper, apart from the typical study of statistical distributions, a number of other dependencies were sought, such as the dependence of the standard deviation of the ship's position in relation to the length of the section, the dependence of ship's size, including draught and width, on the position on the fairway, or very important dependences of the mean and standard deviation of the ship's position in relation to the distance to safe isobaths.

  1. »Conclusions should be clarified and described concisely, moving some experimental results to the Section 3 Results.«

Answer: We have tried our best to consider your suggestion to upgrade the article. We sincerely hope that we have succeeded in improving our work per your expectations.

Yours Sincerely,

dr. Andrej Androjna

Round 2

Reviewer 3 Report

This reviewer finded out that major comment was reflected into the revision. 

Regardless of this, it is desirable that the line no. 75 move to line no.134.

In the introduction, it is not suitable to describe such preliminary study in details including the  model equations regarding lateral distribution of ships.   

Author Response

(The authors gave the same response as above.)
